# The Increase in the Frequency and Amplitude of the Beating of Isolated Mouse Tracheal Cilia Reactivated by ATP and cAMP with Elevation in pH

**DOI:** 10.3390/ijms25158138

**Published:** 2024-07-26

**Authors:** Akari Kobayashi, Kotoku Kawaguchi, Shinji Asano, Hong Wu, Takashi Nakano, Toshio Inui, Yoshinori Marunaka, Takashi Nakahari

**Affiliations:** 1Research Unit for Epithelial Physiology, Research Organization of Science and Technology, Biwako Kusatsu Campus (BKC), Ritsumeikan University, Kusatsu 525-8577, Japan; ikimonomoka0323@gmail.com (A.K.); k-kawagu@dent.meikai.ac.jp (K.K.); ashinji@ph.ritsumei.ac.jp (S.A.); t-inui@saisei-mirai.or.jp (T.I.); marunaka@koto.kpu-m.ac.jp (Y.M.); 2Department of Molecular Physiology, Faculty of Pharmacy, Biwako Kusatsu Campus (BKC), Ritsumeikan University, Kusatsu 525-8577, Japan; 3Department of Microbiology and Infection Control, Faculty of Medicine, Osaka Medical and Pharmaceutical University, 2-7 Daigaku-Machi, Takatsuki 569-8686, Japan; hong.wu@ompu.ac.jp (H.W.); tnakano@ompu.ac.jp (T.N.); 4Saisei Mirai Clinics, Moriguchi 570-0012, Japan; 5Medical Research Institute, Kyoto Industrial Health Association, Kyoto 604-8472, Japan

**Keywords:** intracellular pH, airway cilia, high-speed video microscopy, ciliary beat frequency, ciliary bend amplitude, ciliary wave form

## Abstract

Single cilia, 100 nm in diameter and 10 µm in length, were isolated from mouse tracheae with Triton X-100 (0.02%) treatment, and the effects of pH on ciliary beating were examined by measuring the ciliary beat frequency (CBF) and the ciliary bend distance (CBD—an index of amplitude) using a high-speed video microscope (250 fps). ATP (2.5 mM) plus 8Br-cAMP (10 µM) reactivated the CBF and CBD in the isolated cilia, similar to the cilia of in vivo tracheae. In the reactivated isolated cilia, an elevation in pH from 7.0 to 8.0 increased the CBF from 3 to 15 Hz and the CBD from 0.6 to 1.5 µm. The pH elevation also increased the velocity of the effective stroke; however, it did not increase the recovery stroke, and, moreover, it decreased the intervals between beats. This indicates that H^+^ (pH_i_) directly acts on the axonemal machinery to regulate CBF and CBD. In isolated cilia priorly treated with 1 µM PKI-amide (a PKA inhibitor), 8Br-cAMP did not increase the CBF or CBD in the ATP-stimulated isolated cilia. pH modulates the PKA signal, which enhances the axonemal beating generated by the ATP-activated inner and outer dyneins.

## 1. Introduction

The beating cilia in airways repeat asymmetric movements with an effective stroke and a recovery stroke, which produce an airway surface fluid flow that sweeps away inhaled small particles from the airways [1,2,3]. Previous studies have demonstrated that ATP reactivated the ciliary beating of demembranated cilia isolated from tracheae using Triton-X 100 treatment [1,4], and further addition of cAMP enhanced the ciliary beat frequency (CBF) to the level observed in tracheal ciliary cells [4]. Thus, ciliary beating can be reproduced through the addition of ATP plus cAMP in isolated cilia.

The airway CBF has been shown to be enhanced by many substances, such as cAMP and cGMP, and intracellular ions, such as Ca^2+^, H^+^ and Cl^−^ [3,5,6,7,8]. Among them, H^+^ (pH) is an important ion regulating the airway CBF. The effects of intracellular pH (pH_i_) on CBF have been examined in a perforated model of human tracheobronchial ciliated cells using α-toxin, suggesting that H^+^ (pH_i_) directly acts on the axonemal machinery, possibly the outer dynein arm [7]. However, the contributions of the cellular process to the pH regulation of cilia were unclear because the experiments were performed in cilia with cells, not cell-free cilia. In airway ciliated cells, HCO_3_^−^ transport, which changes pH_i_, plays an important role in maintaining healthy airways [7,8,9]. Moreover, the pH_i_ has also been shown to affect the CBD (ciliary bend distance—an index of amplitude) [6]. Thus, pH_i_ may be an important factor for controlling CBF and CBD in airway ciliary cells [6,7]. In experiments using cell-free cilia, it may be possible to examine the target proteins of pH (H^+^) regulation of CBF and CBD in the axoneme, such as outer dynein arms (ODAs) and inner dynein arms (IDAs).

A cilium has an axonemal structure [3,5,10,11]. The axoneme is composed of a central pair of microtubules surrounded by nine doublet microtubules (A and B tubules). Two molecular motors attached to the A tubule, ODAs and iIDAs, slide the next outer doublet tubules to produce axonemal bending [10,11]. The ODAs control the frequency, and the IDAs control the wave form, including the amplitude [11]. pH may affect the axonemal machinery controlling the ODAs (CBF) and IDAs (the waveform, including the CBD) in airway cilia because ODAs and IDAs have been considered similar in structure and function [11].

In this study, we used demembranated cilia treated with triton-X 100. Stimulation with ATP plus 8Br-cAMP reactivated the beating of the isolated tracheal cilia of mice, as shown in a previous report [4]. We observed the beating of isolated cilia using a video microscope (HSVM) and measured the CBF and CBD [6]. The CBF has already been shown to be modulated by intracellular pH (pH_i_) in perforated airway ciliary cells [7]. In airway ciliary cells, an increase in pH_i_ has been shown to enhance the CBD [6]. It is difficult to fix the pH_i_ of the inside cilia in airway ciliary cells. However, the isolated cilia allowed us to control the pH of the inside cilia. Yet it still remains uncertain whether or not pH directly controls the structural proteins of the axoneme regulating CBF and CBD. The reactivation of the isolated cilia allowed us to examine the effects of pH on the axoneme without any cellular process. In this study, we examined the effects of pH on CBF and CBD in demembranated cilia isolated from mouse tracheae stimulated by ATP plus 8Br-cAMP.

## 2. Results

### 2.1. Effects of ATP on CBF and CBD

Cilia were isolated from mouse tracheae using a Triton-X 100 (0.02%) treatment (the extraction solution). After the isolation of the cilia, the perfusion solution was switched from an extraction solution to an intracellular solution at 37 °C (pH = 7.4; Table 1). An isolated cilium, one end of which was attached to the coverslip, was observed using an HSVM (250 frames per second (fps); please see Appendix A). Video-frame images of an isolated cilium are shown in Figure 1A,B, in which a line (a–b) is set on the cilium. The changes in the light intensity on the line (a–b), which were returned by an image analysis program, are shown in Figure 1C. The changes shown in Figure 1A,B are indicated by the white arrows labeled A and B in Figure 1C. The isolated cilium without any stimulation fluctuated with a small amplitude (Figure 1C).

The addition of ATP (2.5 mM) reactivated the repeated ciliary beating (Figure 1D–F and Appendix A), which was recorded using a different cilium from the cilium shown in Figure 1A,B. Figure 1D,E show the start and end of the effective stroke, respectively, and a line (a–b) is set on the cilium. Figure 1F shows the changes in the light intensity on the line (a–b) returned by the image analysis program. The changes shown in Figure 1D,E are indicated by the white arrows labeled D and E in Figure 1F. The white traces in Figure 1F show two beats of the cilium. ATP (2.5 mM) activated the repeated beating of the isolated cilium.

The concentration effects of ATP on the CBF and CBF in isolated tracheal cilia are shown in Figure 2A. The addition of ATP increased the CBF and CBF in a concentration-dependent manner (Figure 2). In the unstimulated cilia, the CBF and CBD were 0.38 ± 0.40 Hz (*n* = 5) and 0.23 ± 0.04 µm (*n* = 5). The addition of 0.5 mM ATP did not increase the CBF; however, 2.5 mM to 7.5 mM ATP reactivated the repeated beating, as shown in Figure 1D–F, and 10 mM ATP did not reactivate it. The values of the CBF and CBD were 0.69 ± 0.50 Hz (*n* = 4) and 0.30 ± 0.12 µm (*n* = 4) at 0.5 mM, 1.44 ± 0.50 Hz (*n* = 5) and 0.44 ± 0.13 µm (*n* = 4) at 2.5 mM, 0.96 ± 0.30 Hz (*n* = 5) and 0.36 ± 0.10 µm (*n* = 4) at 7.5 mM, and 0.40 ± 0.27 Hz (*n* = 4) and 0.20 ± 0.10 µm (*n* = 4) at 10 mM.

The effects of pH on CBF and CBD were examined in isolated cilia activated by 2.5 mM ATP. The increases in pH from 7.0 to 8.0 did not change the CBF or CBD of the isolated cilia stimulated by 2.5 mM ATP (Figure 2B).

Thus, ATP alone activated the repeated beating of isolated cilia. However, after ATP activation, the CBF and CBD values were much smaller than those observed in tracheal ciliary cells in vivo. However, changes in pH did not affect the CBF and CBD in the isolated cilia stimulated by 2.5 mM ATP.

### 2.2. Repeated Beating of Isolated Cilia Reactivated by ATP Plus 8Br-cAMP

Isolated cilia were stimulated by ATP (2.5 mM) plus 8BrcAMP (10 µM). The switch to the reactivation solution containing ATP plus 8Br-cAMP (pH 7.4; Table 1) enhanced the repeated beating of isolated cilia at 37 °C (Figure 3D–F; see Appendix A). Figure 3D,E show the start and the end of the effective stroke of an isolated cilium stimulated by the reactivation solution. Changes in the light intensity on the line (c–d) in Figure 3D,E are shown in Figure 3F. In this cilium, the CBF was 6–7 Hz and the CBD was 1–1.2 µm.

### 2.3. Effects of pH on the CBF and CBD of Isolated Cilia

After the reactivation of the isolated cilia, the pH of the reactivation solution was decreased by 0.2 steps from 7.4 to 7.0. Then, the pH of the solution was increased by 0.2 steps until it reached 8.0. In some experiments, the pH of the solution was first increased by 0.2 steps to 8.0 and then decreased by 0.2 steps until it reached 7.0. Figure 3 shows the typical responses of an isolated cilium stimulated by the reactivation solutions at pH 7.0 (Figure 3A–C), pH 7.4 (Figure 3D–F) and pH 8.0 (Figure 3G–I). The distance from the peak to the bottom of the beating wave shown in the light intensity change was measured over ten beats, and the average value was used as the CBD. Frame images of an isolated cilium reactivated at pH 7.0 are shown in Figure 3A–C. A video image is shown in Appendix A. Changes in the light intensity on the line (a–b) set on the cilium in Figure 3A,B are shown in Figure 3C. Figure 3A,B show the start and end of the effective stroke in an isolated cilium, which are shown by the arrows marked A and B in Figure 3C. In this cilium, the CBF and CBD were 3 Hz and 0.6 µm at pH 7.0. Frame images of the beating and the beating wave of an isolated cilium at 7.4 are shown in Figure 3D–F (see Appendix A). The details have already been described above. In this cilium, the CBF and CBD were 7 Hz and 1 µm. Frame images of the beating of an isolated cilium at pH 8.0 are shown in Figure 3G,H, which show the start and end of the effective stroke, respectively. A video image is shown in Appendix A. The changes in the light intensity on the line (e–f) set on the cilium in Figure 3G,H are shown in Figure 3I. The changes shown in Figure 3G,H are indicated by the arrows labeled G and H in Figure 3I. In this cilium, the CBF and CBD were 12 Hz and 1.5 µm. As the pH of the solution increased, the CBF and CBD increased, and as the pH decreased, they decreased.

The effects of pH on CBF and CBD are summarized in Figure 4. Isolated cilia were reactivated with ATP plus 8Br-cAMP. The numbers of cilia used for the experiments were as follows: 4 cilia at pH 7.0, 4 cilia at pH 7.2, 11 cilia at pH 7.4, 6 cilia at pH 7.6, 6 cilia at pH 7.8 and 4 cilia at pH 8.0. The CBF (Hz) and CBD (µm) at each pH are shown in Figure 4A. As the pH of the reactivation solution increased from 7.0 to 8.0, the CBF linearly increased from 3.4 Hz to 15.4 Hz. However, the CBD had a different pH dependency. As the pH increased from 7.0 to 7.6, the CBD linearly increased from 0.57 µm to 1.31 µm. However, as the pH increased from 7.6 to 8.0, the CBD did not increase. The CBD at pH 8.0 was 1.45 µm.

We also calculated the ratios of the CBF and CBD, (CBF/CBF_pH 7.4_ and CBD/CBD_pH 7.4_, normalized by the CBF and CBD at pH 7.4) (Figure 4B). The CBF ratios were 0.37 at pH 7.0 (*n* = 4), 0.57 at pH 7.2 (*n* = 4), 1.31 at pH 7.6 (*n* = 6), 1.56 at pH 7.8 (*n* = 6) and 1.81 at pH 8.0 (*n* = 4). Thus, the decrease in the pH from 7.4 to 7.0 decreased the CBF by 60%, and the increase in the pH from 7.4 to 8.0 increased the CBF by 80%. The CBD ratios were 0.60 at pH 7.0 (*n* = 4), 0.83 at pH 7.2 (*n* = 4), 1.29 at pH 7.6 (*n* = 6), 1.34 at pH 7.8 (*n* = 6) and 1.42 at pH 8.0 (*n* = 4). The decrease in the pH from 7.4 to 7.0 decreased the CBD by 40%, and the increase in the pH from 7.4 to 7.6 increased the CBD by 30%; however, further increases in the pH to 7.8 or 8.0 did not increase the CBD.

The different pH dependencies of the CBF and CBD suggest that CBF and CBD are controlled by different PKA signaling pathways, which have different pH dependencies.

### 2.4. Effects of pH on the Beating of Isolated Cilia Reactivated by ATP Plus 8Br-cAMP

#### 2.4.1. Effective Stroke and Recovery Stroke

The CBF and CBD of the isolated cilia reactivated by ATP plus 8Br-cAMP were enhanced by the elevation in pH. We examined the effects of pH on the effective stroke and the recovery stroke in the reactivated isolated cilia. We counted the number of frames from the start to the end of the effective and recovery strokes using the images of the light intensity changes. In each cilium, we counted the number of frames for the effective and recovery strokes over 10 beats at each pH, and the average was used as the duration time (the frame rate was 250 fps, 4 ms/frame). The times for the effective and recovery strokes at each pH are plotted in Figure 5A. The numbers of cilia used for the experiments were *n* = 4 at pH 7.0, *n* = 4 at pH 7.2, *n* = 11 at pH 7.4, *n* = 6 at pH 7.6, *n* = 6 at pH 7.8 and *n* = 4 at pH 8.0. Regarding the effective stroke, the duration time decreased from 136 ms to 44 ms as the pH increased from 7.0 to 7.4, and it did not change (36–37 ms) as the pH increased from 7.6 to 8.0. However, regarding the recovery stroke, the duration time was constant as the pH increased from 7.0 to 8.0 (37–40 ms).

#### 2.4.2. Intervals between Beats

Figure 3 shows that an elevation in pH decreased the intervals between beats. At each pH, the intervals between beats were obtained from three cilia and were measured using 2–4 records (for 4–8 s) for the isolated cilia. As shown in Figure 3, the intervals changed in the cilia with a constant pH (Figure 3). For example, the intervals varied between 584 ms and 112 ms at pH 7.0 and between 8 ms and 72 ms at pH 8.0. The intervals between beats were plotted against the pH (Figure 5B). The intervals depended on the pH; as the pH increased, the intervals decreased. The mean intervals between beats were 234 ms at pH 7.0 (*n* = 60 beats), 212 ms at pH 7.2 (*n* = 60), 120 ms at pH 7.4 (*n* = 82), 53 ms at pH 7.6 (*n* = 92), 44 ms at pH 7.8 (*n* = 63) and 31 ms at pH 8.0 (*n* = 82).

Thus, an increase in pH accelerated the bending speed of the effective stroke, but it did not change the release speed of the recovery stroke. Moreover, it decreased the intervals between beats (the non-beating state in which dyneins may be inactive) in the isolated cilia stimulated by ATP plus 8Br-cAMP.

### 2.5. Effects of ATPγS and PKI on the CBF and CBD of Isolated Cilia Reactivated by ATP or ATP Plus 8Br-cAMP

The following experiments were carried out at pH 7.4. Isolated cilia were stimulated with an analogue of ATP (ATPγS). The addition of ATPγS (2.5 mM) and ATPγS plus 8Br-cAMP (10 µM) did not activate beating of the isolated cilia (Figure 6A).

The isolated cilia were first stimulated by ATP plus 8Br-cAMP. Then, 8Br-cAMP was removed from the reactivation solution 5 min after the reactivation. However, the beating continued without any decreases in the CBF or CBD. We examined the effects of 1 µM PKI-amide (a PKA inhibitor) on isolated cilia stimulated by ATP plus 8Br-cAMP. Prior treatment with PKI-amide for 10 min and then stimulation with ATP plus 8Br-cAMP did not reactivate the beating of isolated cilia (Figure 6B). However, after the reactivation of isolated cilia by ATP plus 8Br-cAMP, the subsequent addition of PKI-amide did not inhibit the beating of isolated cilia (Figure 6B)

## 3. Discussion

The present study demonstrated that an increase in pH enhances the CBF and CBD in isolated tracheal cilia (cell-free axonemes) of mice which have been reactivated by ATP plus 8Br-cAMP. There are reports showing pH control of ciliary beating in airway ciliary cells [6,7] and in sperm [12,13]. The increases in the CBF and CBD observed in cell-free axonemes indicate that pH is directly associated with the axonemal proteins activated by c-AMP. Cyclic AMP-dependent phosphorylation is known to activate ciliary beating [14]. Previous studies have demonstrated that the c-AMP-dependent phosphorylation of axonemal proteins activates flagellar beating [12,15,16] and that the activity of dynein ATPase is sensitive to pH; it was 3.5-fold higher at pH 7.6 than at pH 7.0 [13,15].

As the pH increased from 7.0 to 8.0, the CBF linearly increased in the isolated tracheal cilia stimulated by ATP plus 8Br-cAMP. However, the increase in the CBD showed a different pH dependency. As the pH increased from 7.0 to 7.6, the CBD linearly increased; however, at a pH ranging from 7.6 to 8.0, the CBD did not increase and remained constant. The different pH dependencies of CBF and CBD indicate that the pH control of CBF and CBD is mediated by the different signaling pathways stimulated by PKA. Studies in Chlamydomonas mutants suggest that ODA regulates the CBF, whereas IDA controls the waveform, including the CBD [11]. The central pairs and radial spokes in the axoneme have been shown to contain several A-kinase anchoring proteins (AKAPs), which are a family of proteins that target PKA to specific intracellular sites and interact with the following PKA regulatory subunits: AKAP240 (240-kDa AKAP), located in the central pair apparatus, and AKAP97 (97-kDa AKAP), located at the base of the radial spoke stalk adjacent to the outer doublet microtubules [16]. The radial spokes (RSs), which repeat as a set of triplets (RS1, RS2 and RS3) every 96 nm along the length of the A tubule, stimulate sliding velocities through the control of dyneins [16,17,18]. The PKA signals from the central pair apparatus transmit to the IDA and then to the ODA through RS1 and to the ODA through RS2 [16]. pH may modulate two PKA signaling pathways through RS1 and RS2 [16,17].

The present study demonstrated that isolated cilia were activated by ATP to produce the repeated beating with a low CBF and a small CBD. Previous reports also demonstrated that ATP activates repeated beating of cilia isolated from mouse tracheae through conformational changes in the dyneins accompanied by ATP hydrolysis [1,4]. The present study demonstrated that a stable analogue of ATP, ATPγS, did not reactivate beating of the isolated cilia. This indicates that the energy produced by the dehydration of ATP is essential to activate ciliary beating.

A previous study demonstrated that c-AMP enhanced the CBF in isolated bovine ciliary axonemes reactivated by ATP [4]. As mentioned above, AKAPs targeting PKA exist in the axoneme. This study demonstrated that 8Br-cAMP enhances both the CBF and CBD in isolated cilia stimulated by ATP and that prior treatment with PKI-amide inhibits the enhancements stimulated by 8Br-cAMP. These findings indicate that the PKA-stimulated phosphorylation of target proteins (including IDAs and ODAs) existing in the axoneme is an important factor to induce repeated beating. Moreover, once the repeated beating was reactivated by 8Br-cAMP, the further addition of PKI-amide or the removal of 8Br-cAMP did not suppress the repeated beating. The activities of phosphatase also appear to be important to maintaining adequate ciliary beating.

The present study demonstrated that the duration time of the effective stroke depends on pH but the recovery stroke does not. For ciliary bending, the dyneins on all doublets are not active at the same time. The switching mechanism controls the effective and recovery strokes; the dyneins on one side of the axoneme axis generate the forward bending (the effective stroke), and the dyneins on the other side of the axoneme axis produce the reverse bending (the recovery stroke) [16,19]. The first switch in the switching mechanism stimulates the dyneins controlling the forward bending, and the second switch stimulates the dyneins controlling the reverse bending, suppressing the dyneins activated by the first switch. An increase in pH enhances the sliding velocity of the dyneins that generate the forward bending activated by the first switch but not the sliding velocity of the dyneins that produce reverse bending.

The intervals between beats also decreased with the increase in pH. In the interval, all dyneins may be inactive. The outer dynein arms showed two different conformations depending on the presence or absence of ATP [10,20]. The dyneins adopt two functional states (active and inactive), and the activation of dyneins requires ATP binding [10]. The ATP binding site of dynein may be sensitive to pH; a low pH may decrease the probability of ATP binding, and a high pH may increase the probability. Further studies are required. The binding of ATP to PKA-stimulated ODAs and IDAs, leading to changes in the interval, may be affected by pH, which modulates the PKA-dependent pathway controlling ODAs and IDAs. At present, we do not know the mechanism controlling the interval between beats.

Stimulation with ATP plus 8Br-cAMP reactivated the isolated cilia in this study. However, only a small number of isolated cilia were reactivated by ATP or ATP plus 8Br-cAMP. The concentrations of triton-X 100 examined were 0.1%, 0.05%, 0.02% and 0.01%. Isolated cilia treated with 0.02% triton-X 100 were reactivated by ATP plus 8Br-cAMP. We examined the structure of the isolated cilia treated with 0.02% triton-X 100 using conventional electron microscopy (Appendix A). Only a small number of isolated cilia retained the 9 + 2 axonemal structure; in most of the isolated cilia, the axonemal structures were not detected. Treatment with triton-X 100 may injure the axonemal structure. Improving the isolation procedure, for instance, by adjusting the concentration of triton-X 100 or the treatment time, may increase the number of isolated cilia with the fine 9 + 2 structure.

## 4. Materials and Methods

### 4.1. Ethical Approval

The experiments were approved by the Committee for Animal Research of Ritsumeikan University BKC (BKC-HM-2017-050). The animals were cared for, and the experiments were carried out according to the guidelines of the committee.

### 4.2. Solutions and Chemicals

The solutions used are summarized in Table 1. The pHs of the control and the extraction solutions were adjusted to 7.4 by adding an appropriate amount of 0.1 M NaOH, and those of the intracellular and reactivation solutions were adjusted to 7.4 by adding an appropriate amount of 0.1 M KOH. The pHs of the reactivation solution were varied from 7.0 to 8.0 by adding 0.1 M HCl or 0.1 M KOH as appropriate.

Triton-X 100, β-mercaptoethanol and dithiothritol were purchased from Fujifilm-Wako (Osaka, Japan), and ATP, ATPγS, 8Br-cAMP and PKI 14–22 amide myristoylated (PKI-A) were purchased from Sigma-Aldrich (St Louis, MO, USA).

### 4.3. Mice and Tracheal Preparation

Female mice (C57BL/6J, 5 weeks of age) were purchased from Shimizu Experimental Animals (Kyoto, Japan), fed standard pellet food and given water ad libitum, and used for experiments at 6–10 weeks of age. The number of mice used for the experiments was 55. First, the mice were anesthetized using inhalational isoflurane (3%), then they were sacrificed by an intraperitoneal injection of a high dose of pentobarbital sodium (100 mg/kg).

After the sacrifice, tracheae were removed from the mice. The connective tissues were removed from the tracheae using fine forceps and scissors. The tracheae were opened by a median incision, and the surface mucous layers were flushed with the control solution five times using a syringe (2.5 mL). After washing the surface mucous layers, the opened tracheae were kept in cooled control solution (4 °C) until the experiments.

### 4.4. Cilia Isolation

A schematic diagram of the experimental set-up depicting the beating of isolated cilia and the method used to measure CBF and CBD is shown in Figure 7.

Each trachea was cut into small pieces (2–3 mm blocks). Two or three tracheal blocks were set in the perfusion chamber (20 µL) containing the control solution (Figure 7, Left panel). The perfusion chamber was set on the stage of the HSVM. The tracheal blocks were perfused with the control solution (37 °C) for 5 min at a constant rate (300 µL/min). Then, the control solution was switched to the extraction solution. After replacing the control solution of the chamber with the extraction solution, the perfusion was stopped. The tracheal blocks in the chamber were incubated with the extraction solution for 2–3 min; then, the coverslip on the chamber was gently tapped using fine forceps several times to isolate the cilia from the tracheal surface, and the trachea was further incubated in the extraction solution at 37 °C for 5 min to attach one end of the isolated cilia to the bottom coverslip of the chamber. After confirming the firm attachment of the isolated cilia using the HSVM, the perfusion solution was switched to the intracellular solution and the isolated cilia were perfused at a constant rate (300 µL/min) at 37 °C (Figure 7, Left panel).

### 4.5. Measurements of CBF and CBD

Each cilium, one end of which was attached to the coverslip, was observed using a high-speed video microscope (HSVM—an inverted microscope (T-2000, Nikon, Tokyo, Japan)) and a high-speed camera (IDP-Express R2000; Photoron, Tokyo, Japan). Isolated cilia were reactivated using the reactivation solution containing ATP (2.5 mM) or ATP (2.5 mM) plus 8Br-cAMP (10 µM) (Table 1). We carefully searched the reactivated cilia, as only small numbers of the isolated cilia were reactivated. A beating cilium was recorded for 2 s at 250 fps using the HSVM. After the experiments, we measured the CBF and CBD using an image analysis program (DippMotionV2D; DITECT, Tokyo, Japan) [6]. We set a line on a beating cilium in the video image, and the program returned the wave of the light intensity changes. The number of peaks was counted for 2 s, and the distances (μm) from the top of the peaks to the bottom were measured over 10 waves (Figure 7, Right panel). The number of peaks and the distances averaged over 10 waves were used to determine the CBFs and CBDs. To examine the effects of the pH on the CBFs and CBDs, the ratios of the CBFs and CBDs (CBF/CBF_pH=7.4_ and CBD/CBD_pH=7.4_) at each pH were calculated. The CBFs and CBDs were measured using 3–6 isolated cilia obtained from at least 3 animals.

The pHs of the reactivation solution were adjusted to 7.0, 7.2, 7.6, 7.8 and 8.0 by adding 1N-HCl or 1N-KOH as appropriate. After the addition of ATP or ATP plus 8Br-cAMP, the switch to the reactivation solution (pH 7.4; Table 1), we searched the isolated cilia reactivated in the perfusion chamber, the ends of which were attached to the coverslips. The beating of the reactivated isolated cilia was observed in the reactivation solution with a pH of 7.4 by the HSVM. The pH of the chamber was changed by switching to the reactivation solution with various pHs (7.0–8.0). The pH of the reactivation solution was decreased by 0.2 steps from 7.4 to 7.0. Then, the pH was increased by 0.2 steps from 7.0 to 8.0. In some experiments, the pH was increased from 7.4 to 8.0 by 0.2 steps and then decreased to 7.4 to 7.0 by 0.2 steps. We repeated the experiments, changing the pH. However, in some cases, the ciliary beating suddenly stopped during the change in pH, or the isolated cilia were left on the coverslips during perfusion. In these cases, we searched the new beating cilia in the chamber and started new experiments using the newly isolated cilia. After the isolation of cilia, the experiments were carried out for 90 min.

### 4.6. Electron-Microscopic Examination

Tracheae were taken from three mice. After removing the mucous layer through flushing, each trachea was opened by a median incision and cut into eight pieces (2–4 mm blocks). The tracheal blocks were vortexed for 2 min in 500 µL of the extraction solution. The samples were centrifuged at 1500× *g* for 2 min. The supernatant containing the cilia was centrifuged twice at 12,000× *g* for 5 min. Then, the cilia were immediately fixed in glutaraldehyde (1%) in 0.15 M phosphate-buffered saline (pH 7.2) at 4 °C for 1 h. The fixed cilia were processed according to the routine procedure for electron microscopy.

### 4.7. Stastistical Analysis

The statistical analysis of the data was carried out using paired and unpaired *t*-tests or one-way ANOVA as appropriate.

## Figures and Tables

**Figure 1 ijms-25-08138-f001:**
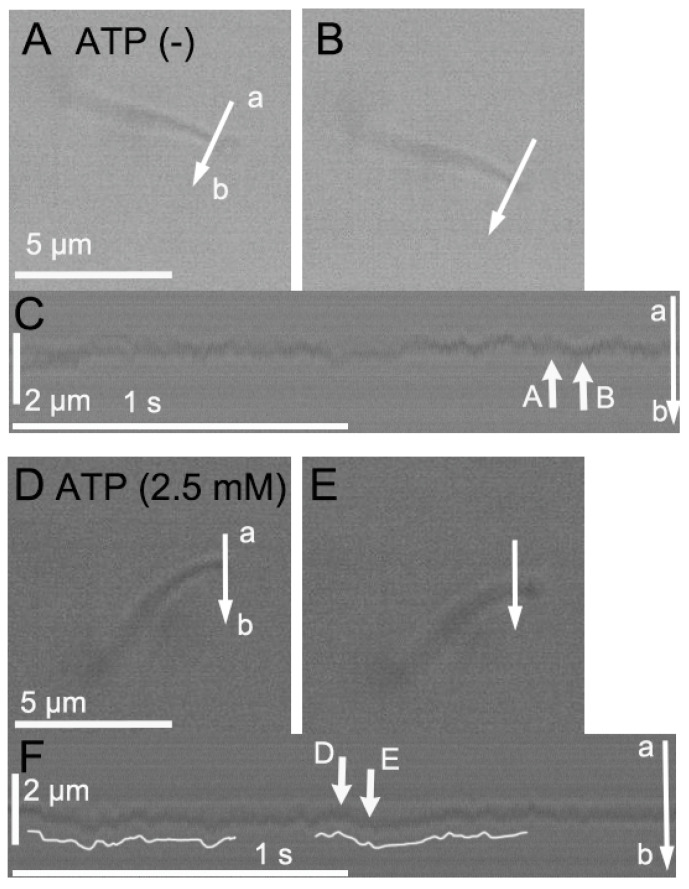
Video-frame images of isolated cilia. Isolated cilia in the intracellular solution (pH 7.4) were observed by an HSVM (250 fps). (**A**–**C**) An isolated cilium before ATP stimulation. The isolated cilium showed fluctuation but no repeated beating. Panels (**A**,**B**) show a small fluctuation in the cilium. Panel (**C**) shows the light intensity change on the line a–b in the panels (**A**,**B**). A small fluctuation in the cilium was detected. The changes shown in panels (**A**,**B**) are indicated by the white arrows labeled A and B in panel (**C**). (**D**–**F**) An isolated cilium reactivated by ATP (2.5 mM). ATP stimulation reactivated the repeated beating of the cilium. Panels (**D**,**E**) show the start and end of the effective forward stroke, respectively. Panel (**F**) shows the light intensity change on the line a–b in panel (**D**). The changes shown in panels (**D**,**E**) are indicated by the white arrows labeled D and E in panel (**F**).

**Figure 2 ijms-25-08138-f002:**
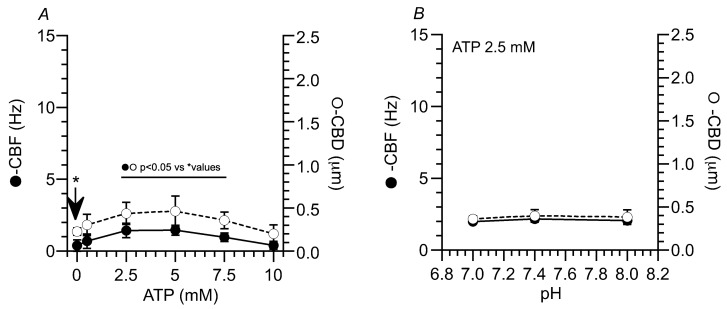
(**A**) The effects of ATP concentration on CBF and CBD in isolated cilia. ATP increased the CBF and CBD in a concentration-dependent manner. As the ATP concentration increased from 0 mM to 5 mM, the CBF and CBD increased; however, at ATP concentrations higher than 7.5 mM, the CBF and CBD decreased. The CBF and CBD showed a similar ATP concentration dependency. (**B**) Effects of pH on the CBF and CBD in isolated cilia reactivated by ATP (2.5 mM). In isolated cilia activated by 2.5 mM ATP, the CBF and CBD were not changed by the elevation in pH from 7.0 to 8.0. The arrow shows the CBF and CBD of cilia without ATP stimulation. The CBFs and CBDs with ATP were compared with * values.

**Figure 3 ijms-25-08138-f003:**
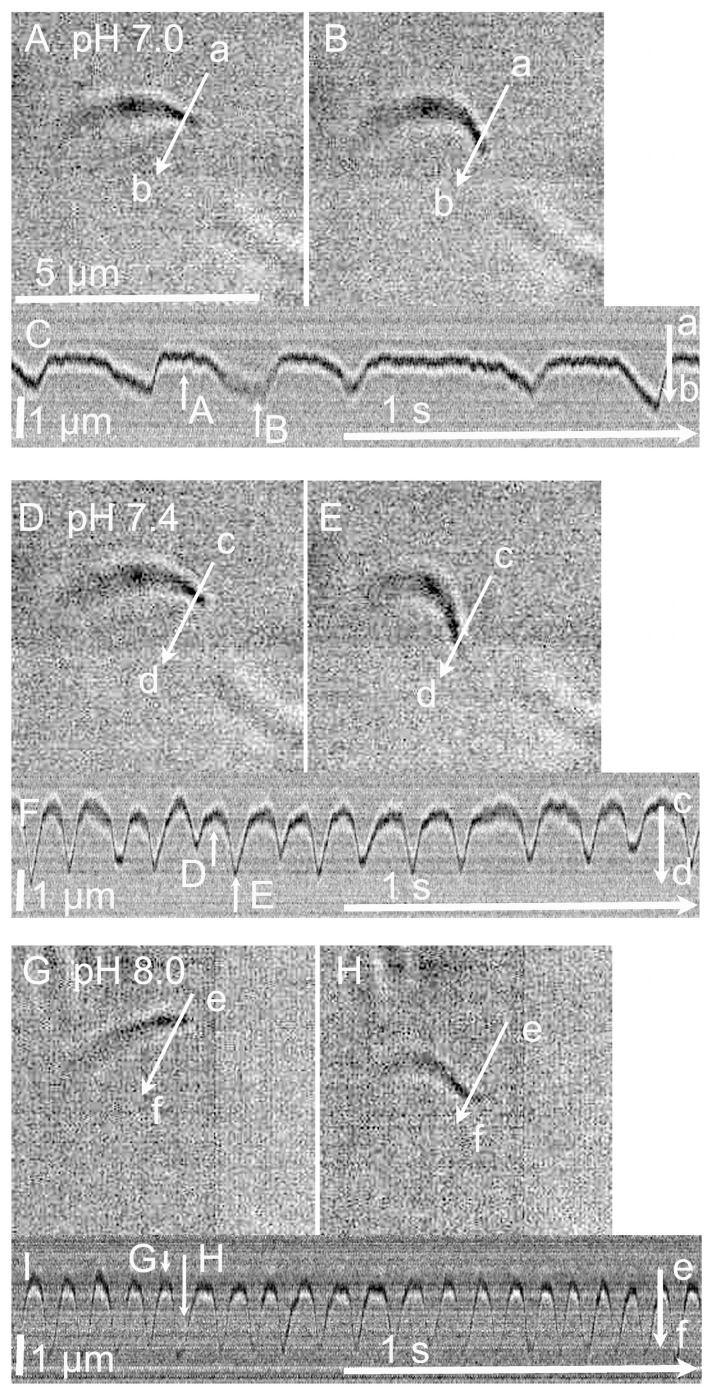
Effects of pH on isolated cilia reactivated by ATP (2.5 mM) plus 8Br-cAMP (10 µM). The addition of ATP plus 8Br-cAMP reactivated the repeated beating. As the pH increased from 7.0 to 8.0, the CBF and CBD increased in the isolated cilia reactivated by ATP plus 8Br-cAMP. (**A**–**C**) pH 7.0. Panels (**A**,**B**) show the start and end of the effective stroke. Panel (**C**) shows the changes in the light intensity on the line (a–b) marked on the cilium in the frame images (panels (**A**,**B**)). The changes shown in panels (**A**,**B**) are indicated by the white arrows A and B, respectively. (**D**–**F**) pH 7.4. Panels (**D**,**E**) show the start and end of the effective stroke. Panel (**F**) shows the changes in the light intensity on the line (c–d) marked on the cilium in the frame images (panels (**D**,**E**)). The changes shown in panels (**D**,**E**) are indicated by the white arrows D and E. (**G**–**I**) pH 8.0. Panels (**G**,**H**) show the start and end of the effective stroke. Panel (**I**) shows the changes in the light intensity on the line (e–f) marked on the cilium in the frame images (panels (**G**,**H**)). The changes shown in panels (**G**,**H**) are indicated by the white arrows G and H. The CBF and CBD of the isolated cilia reactivated by ATP plus 8Br-cAMP were enhanced by the increase in pH.

**Figure 4 ijms-25-08138-f004:**
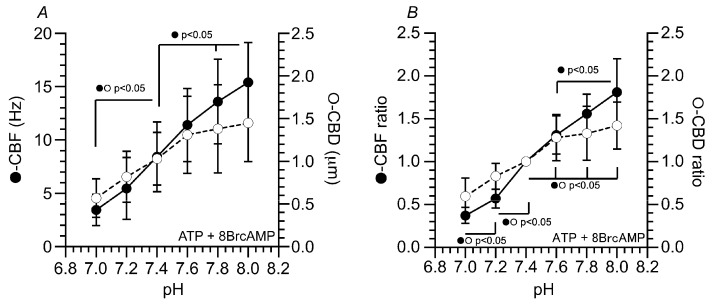
Effects of pH on the CBF and CBD in isolated cilia reactivated by ATP plus 8Br-cAMP. (**A**) Changes in CBF (Hz) and CBD (µm). In isolated cilia reactivated by ATP plus 8Br-cAMP, as the pH increased from 7.0 to 8, the CBF linearly increased. However, the pH dependency of the CBD increase was different from that of the CBF increase. As the pH increased from 7.0 to 7.6, the CBD linearly increased, but as the pH increased from 7.6 to 8.0, the CBDs were almost constant (no increase). (**B**) Changes in the CBF ratio and CBD ratio. CBF and CBD were normalized by CBF_pH=7.4_ and CBD_pH=7.4_.

**Figure 5 ijms-25-08138-f005:**
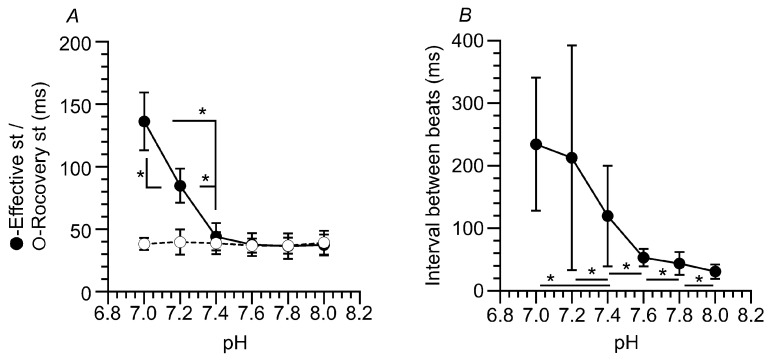
(**A**) Effects of pH on the time for the effective and recovery strokes in isolated cilia reactivated by ATP plus 8Br-cAMP. The time for the effective stroke linearly decreased as the pH increased from 7.0 to 7.4 and did not change as the pH increased from 7.4 to 8.0. Changes in pH from 7.0 to 8.0 did not affect the time for the recovery stroke. (**B**) Effects of pH on the intervals between beats. As the pH increased, the intervals decreased. * Significantly different (*p* < 0.05).

**Figure 6 ijms-25-08138-f006:**
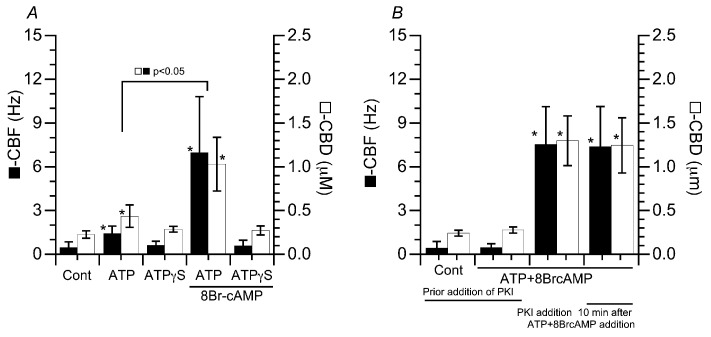
Effects of ATPγS (an ATP analogue) and PKI-amide (a PKA inhibitor) on the CBF and CBD in isolated cilia reactivated by ATP or ATP plus 8Br-cAMP. (**A**) ATPγS. The addition of ATPγS did not reactivate the repeated beating of isolated cilia. ATPγS plus 8Br-cAMP did not increase the CBF or CBD in isolated cilia. (**B**) PKI-amide. Prior treatment with PKI-amide (1 µM) did not increase the CBF or CBD in isolated cilia stimulated by ATP plus 8Br-cAMP. When cilia were reactivated by ATP plus 8Br-cAMP, the further addition of PKI-amide did not decrease CBF or CBD. * Significantly different vs. control (*p* < 0.05).

**Figure 7 ijms-25-08138-f007:**
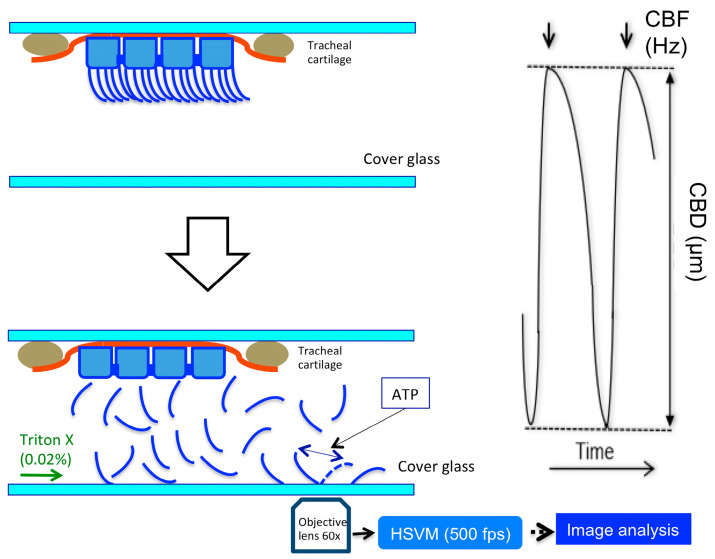
Schematic diagram of the experimental set-up. (**Left panel**) A tracheal block was set in the perfusion chamber. To isolate the cilia, the perfusion solution was switched to the extraction solution. After the isolation of the cilia, the extraction solution was switched to the intracellular solution. The isolated cilia were stimulated by the reactivation solution. The beating of cilia was observed using an HSVM. The beating cilia were recorded, and after the experiments the CBF and CBD were measured from the recorded images. (**Right panel**) We set lines on the cilia in the recorded video images; the light intensity changes on the lines were calculated using an image analysis program. CBF and CBD were measured from the images of light intensity change.

**Table 1 ijms-25-08138-t001:** Composition of solutions.

Solution	Control Solution	Extraction Solution	Intracellular Solution	Reactivation Solution
NaCl (mM)	50	50	0	0
KCl (mM)	0	0	50	50
CaCl_2_ (mM)	10	10	0	0
MgSO_4_ (mM)	0	0	4	4
EDTA (mM)	0.5	0.5	0.5	0.5
HEPES (mM)	20	20	20	20
β-mercaptoethanol (mM)	7	7	0	0
Dithiothritol (mM)	5	1	1	1
Triton X-100 (%)	0	0.02	0	0
ATP (mM)	0	0	0	2.5
8Br-cAMP (µM)	0	0	0	10

## Data Availability

The data that support the findings of this study are in the paper itself; no shared data were used in the paper.

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
