# Peer review of "The Increase in the Frequency and Amplitude of the Beating of Isolated Mouse Tracheal Cilia Reactivated by ATP and cAMP with Elevation in pH"

_ijms, 2024, doi:10.3390/ijms25158138_

Round 1
Reviewer 1 Report
Comments and Suggestions for Authors
In this paper, the authors set out to examine and describe to effects of pH on CBF and CBD. Airway cilia function is crucial in pulmonary clearance of xenogens from the airways, and greater understanding of the mechanisms involved in this process are important in understanding pathogenesis of pulmonary disease. The approach used by the authors to study pH effects on cilia is unique and I applaud the authors on their efforts put forth in this paper. The manuscript was organized, well-written, and a pleasure to review.
Comments:
1. The authors use the word “flame” to describe the images of the cilia, I am not sure if this is jargon but I am unfamiliar with the term. A sentence to describe what a “flame” is in terms of this study would be helpful.
2. Figure 1. Frame (E) looks identical to (D) and is not reflective of the change shown in frame (F). This needs to be corrected.
3. Figure 4. In reference to “In isolated cilia reactivated by ATP alone, CBF and CBD were not enhanced by the elevation in the pH.” The response is difficult to see in that figure (A). The diamond symbols are smaller than the circles, and the text “8BrcAMP(+)” needs to be lowered closer to the graphed lines. I appreciate the authors attempt to make less figure panes, but I highly suggest that the authors consider making that ATP(+/-) response a separate figure pane because the figure is too busy and takes away from the point the authors are trying to make.
4. Figure 5. Figure legend needs to include what asterisks mean in this figure, as was done in figure 6.
Author Response
IN THIS PAPER, THE AUTHORS SET OUT TO EXAMINE AND DESCRIBE TO EFFECTS OF PH ON CBF AND CBD. AIRWAY CILIA FUNCTION IS CRUCIAL IN PULMONARY CLEARANCE OF XENOGENS FROM THE AIRWAYS, AND GREATER UNDERSTANDING OF THE MECHANISMS INVOLVED IN THIS PROCESS ARE IMPORTANT IN UNDERSTANDING PATHOGENESIS OF PULMONARY DISEASE. THE APPROACH USED BY THE AUTHORS TO STUDY PH EFFECTS ON CILIA IS UNIQUE AND I APPLAUD THE AUTHORS ON THEIR EFFORTS PUT FORTH IN THIS PAPER. THE MANUSCRIPT WAS ORGANIZED, WELL-WRITTEN, AND A PLEASURE TO REVIEW.
COMMENTS:
- THE AUTHORS USE THE WORD “FLAME” TO DESCRIBE THE IMAGES OF THE CILIA, I AM NOT SURE IF THIS IS JARGON BUT I AM UNFAMILIAR WITH THE TERM. A SENTENCE TO DESCRIBE WHAT A “FLAME” IS IN TERMS OF THIS STUDY WOULD BE HELPFUL.
I am sorry. The word “flame” is wrong. The word “frame” is correct.
I corrected to the “frame” throughout the manuscript
- FIGURE 1. FRAME (E) LOOKS IDENTICAL TO (D) AND IS NOT REFLECTIVE OF THE CHANGE SHOWN IN FRAME (F). THIS NEEDS TO BE CORRECTED.
I have corrected the frame image in Fig. 1E.
- FIGURE 4. IN REFERENCE TO “IN ISOLATED CILIA REACTIVATED BY ATP ALONE, CBF AND CBD WERE NOT ENHANCED BY THE ELEVATION IN THE PH.” THE RESPONSE IS DIFFICULT TO SEE IN THAT FIGURE (A). THE DIAMOND SYMBOLS ARE SMALLER THAN THE CIRCLES, AND THE TEXT “8BRCAMP(+)” NEEDS TO BE LOWERED CLOSER TO THE GRAPHED LINES. I APPRECIATE THE AUTHORS ATTEMPT TO MAKE LESS FIGURE PANES, BUT I HIGHLY SUGGEST THAT THE AUTHORS CONSIDER MAKING THAT ATP(+/-) RESPONSE A SEPARATE FIGURE PANE BECAUSE THE FIGURE IS TOO BUSY AND TAKES AWAY FROM THE POINT THE AUTHORS ARE TRYING TO MAKE.
I have added the figure 2B showing the pH effects of isolated cilia reactivated by ATP alone. I have removed the plots showing the pH effects of isolated cilia reactivated by ATP alone from Fig. 4A.
- FIGURE 5. FIGURE LEGEND NEEDS TO INCLUDE WHAT ASTERISKS MEAN IN THIS FIGURE, AS WAS DONE IN FIGURE 6.
I have added the “*significantly different” in Figure 5 legend.
Reviewer 2 Report
Comments and Suggestions for Authors
Abstract:
Insufficient information regarding the procedures and outcomes is presented in the abstract. There should be more details included, like sample size, methods of measurement, important discoveries, and conclusions.
Introduction:
- The introduction gives useful background information on pH effects and airway cilia.
To strengthen it, though, the following would be helpful: - Giving a more detailed description of the research gap or open question being addressed;
- Giving additional context regarding the potential implications/applications of the research;
- introduce the potential genetic mechanisms involved in respiratory cilia dysfunction, type II inflammation. discuss and cite doi:10.1111/coa.13870
- Reducing the level of detail regarding axonemal structure;
- Focusing more on what is immediately relevant.
Methods:
- More information is required regarding the sample preparation, such as the quantity of mice utilized, the method of trachea harvesting, and the isolation of cilia.
"- There are no precise measurements for pH modulation, CBD, or CBF. The process of taking and analyzing the measurements has to be covered in more detail.
Details regarding the statistical analysis of the data are required.
Results:
- The data demonstrating the effects of pH on CBF and CBD is good. The way the data is presented, though, might be more appealing.
- Repetitive and redundant panels that could be combined can be found in Figures 1 and 3.
- Sample sizes vary widely throughout groups and are tiny. Stronger findings would result from larger, more reliable sample sizes.
The significance of detected discrepancies must be ascertained by statistical analysis.
"- The main conclusions depicted in the figures require further explanation in the text.
Discussion:
- The discussion provides good context about how pH may be regulating dynein arms and beating.
"- It may be improved, though, by concentrating more on how to interpret their findings rather than providing background information on the structure and function of cilia.
- it should be useful analyze and correlate the findings with clinical outcomes as voice rehabilitation. discuss and cite doi:10.1016/j.jvoice.2021.09.040
- A discussion of the study's shortcomings and potential future paths is required.
Comments on the Quality of English Languageany
Author Response
ABSTRACT:
INSUFFICIENT INFORMATION REGARDING THE PROCEDURES AND OUTCOMES IS PRESENTED IN THE ABSTRACT. THERE SHOULD BE MORE DETAILS INCLUDED, LIKE SAMPLE SIZE, METHODS OF MEASUREMENT, IMPORTANT DISCOVERIES, AND CONCLUSIONS.
I rewrote the abstract according to the reviewer’s comments
INTRODUCTION:
- THE INTRODUCTION GIVES USEFUL BACKGROUND INFORMATION ON PH EFFECTS AND AIRWAY CILIA.
TO STRENGTHEN IT, THOUGH, THE FOLLOWING WOULD BE HELPFUL: - GIVING A MORE DETAILED DESCRIPTION OF THE RESEARCH GAP OR OPEN QUESTION BEING ADDRESSED;
- GIVING ADDITIONAL CONTEXT REGARDING THE POTENTIAL IMPLICATIONS/APPLICATIONS OF THE RESEARCH;
I changed the 2nd paragraph of the introduction as follows:
“The airway CBF has been shown to be enhanced by many substances, such as cAMP and cGMP, and intracellular ions, such as Ca2+, H+ and Cl- [3, 5-8]. Among them, H+ (pH) is an important ion regulating the airway CBF. The effects of the intracellular pH (pHi) on CBF have been examined in a perforated model of human tracheobronchial ciliated cells using a-toxin, suggesting that H+(pHi) directly acts on the axonemal machinery, possibly the outer dynein arm [7]. However, contributions of the cellular process on the pH regulation of cilia were unclear. Because the experiments were performed in the cilia with cell, not cell-free cilia. In airway ciliated cells, the HCO3- transport, which changes pHi, plays an important role in maintaining healthy airways, [7, 8]. Moreover, it has also been shown to affect the CBD (ciliary bend distance, an index of amplitude) [6]. Thus, the pHi may be an important factor for controlling CBF and CBD in airway ciliary cells [6, 7]. In the experiments using the cell-free cilia, it may be possible to examine the target proteins of pH (H+) regulating CBF and CBD in the axoneme, such as ODA or IDA.”
- INTRODUCE THE POTENTIAL GENETIC MECHANISMS INVOLVED IN RESPIRATORY CILIA DYSFUNCTION, TYPE II INFLAMMATION. DISCUSS AND CITE DOI:10.1111/COA.13870
There are many genetic disorders impairing the ciliary beating. Moreover, type II inflammation in the chronic rhinosinusitis (CRS) induce ciliary dysfunction in nasal or sinus ciliary beating. The review introduced by the reviewer (DOI:10.1111/COA.13870), the single-nucreotide polymorphism (SNP) plays a key role in CRS genetic predisposition. The SNP affects Ion channels (K+, Cl-, CFTR), receptors) leading to the polyp formation. However, there is no evidence showing that these factors affect axoneme. These ion channels and receptors exist in the cellular membrane, not in the axoneme. I used the demembranated axoneme. We focused on the effects of pH on the axoneme in this study. It is difficult to discuss genetic mechanism in respiratory ciliary dysfunction. I have no idea to discuss the genetic mechanism in the pH regulation of the axonemal beating.
- REDUCING THE LEVEL OF DETAIL REGARDING AXONEMAL STRUCTURE;
I changed the 2nd paragraph of the introduction as follows:
“A cilium has an axonemal structure [3, 5, 10, 11]. The axoneme is composed of a central pair microtubules surrounded by nine doublet microtubules (A and B tubules). Two molecular motors attached to the A-tubule, outer dynein arms (ODAs) and inner dynein arms (IDAs), slide the next outer doublet tubules to produce the axonemal bending [10, 11]. The ODAs control the frequency, and the IDAs control the wave form including the amplitude [11]. The pH may affect the axonemal machinery controlling the ODAs (CBF) and IDAs (waveform including CBD) in airway cilia, because the ODA and IDA have been considered similar in structure and function [11].”
- FOCUSING MORE ON WHAT IS IMMEDIATELY RELEVANT.
I changed the 2nd paragraph of the introduction as follows:
“In this study, we used the cilium demembranated using a treatment of triton-X 100. Stimulation with ATP plus 8Br-cAMP reactivated the beating in the isolated tracheal cilium of mice, as shown in a previous report [4]. We observed the beating of isolated cilium using a high-speed video-microscope (HSVM) and measured the CBF and CBD [6]. The CBF has already been shown to be modulated by intracellular pH (pHi) in perforated airway ciliary cells [7]. In airway ciliary cells, an increase in pHi has been shown to enhance the CBD [6]. It is difficult to fix the pHi of the inside cilia in airway ciliary cells. However, the isolated cilia allow us to control the pH of the inside cilia. However, it still remains uncertain whether or not the pH directly controls the structural proteins of the axoneme regulating CBF and CBD. The isolated cilia reactivated allows us to examine the effects of pH on the axoneme without any cellular process. In this study, we examined the effects of the pH on the CBF and CBD in the cilia (demembranated) isolated from mice trachea stimulated by ATP plus 8Br-cAMP.”
METHODS:
- MORE INFORMATION IS REQUIRED REGARDING THE SAMPLE PREPARATION, SUCH AS THE QUANTITY OF MICE UTILIZED, THE METHOD OF TRACHEA HARVESTING, AND THE ISOLATION OF CILIA.
"- THERE ARE NO PRECISE MEASUREMENTS FOR PH MODULATION, CBD, OR CBF. THE PROCESS OF TAKING AND ANALYZING THE MEASUREMENTS HAS TO BE COVERED IN MORE DETAIL.
DETAILS REGARDING THE STATISTICAL ANALYSIS OF THE DATA ARE REQUIRED.
I added the information regarding the sample preparation and the more detailed information about the CBF and CBd measurements.
I added the section of the statistical analysis
(see methods).
RESULTS:
- THE DATA DEMONSTRATING THE EFFECTS OF PH ON CBF AND CBD IS GOOD. THE WAY THE DATA IS PRESENTED, THOUGH, MIGHT BE MORE APPEALING.
- REPETITIVE AND REDUNDANT PANELS THAT COULD BE COMBINED CAN BE FOUND IN FIGURES 1 AND 3.
The frame images and the light intensity changes were shown in Fig. 1 and 3. Fig. 1 shows the effects of ATP on the isolated cilia. Fig. 3 shows the effects of ATP plus 8Br-cAMP. I think these panels are necessary to show the reactivated beating of isolated cilia.
The combine of Fig. 1 and Fig 3 is not difficult, but the meanings of two figures are different. I would like to show Fig. 1 and Fig. 3 separately.
- SAMPLE SIZES VARY WIDELY THROUGHOUT GROUPS AND ARE TINY. STRONGER FINDINGS WOULD RESULT FROM LARGER, MORE RELIABLE SAMPLE SIZES.
One experiment was carried out using tracheae obtained from at least 3 animals. We show the all data obtained. I am sorry to add the new data.
THE SIGNIFICANCE OF DETECTED DISCREPANCIES MUST BE ASCERTAINED BY STATISTICAL ANALYSIS.
"- THE MAIN CONCLUSIONS DEPICTED IN THE FIGURES REQUIRE FURTHER EXPLANATION IN THE TEXT.
I added the statistical analysis and the conclusion of figures in the text.
DISCUSSION:
- THE DISCUSSION PROVIDES GOOD CONTEXT ABOUT HOW PH MAY BE REGULATING DYNEIN ARMS AND BEATING.
"- IT MAY BE IMPROVED, THOUGH, BY CONCENTRATING MORE ON HOW TO INTERPRET THEIR FINDINGS RATHER THAN PROVIDING BACKGROUND INFORMATION ON THE STRUCTURE AND FUNCTION OF CILIA.
I changed the discussion.
- it should be useful analyze and correlate the findings with clinical outcomes as voice rehabilitation. discuss and cite doi:10.1016/j.jvoice.2021.09.040
I am sorry again. I do not have any idea for the relationship between voice rehabilitation and the pH regulation of axoneme.
- A DISCUSSION OF THE STUDY'S SHORTCOMINGS AND POTENTIAL FUTURE PATHS IS REQUIRED.
I added the discussion about above.